# Investigations of Hydrodynamic Force Generated on the Rotating Cylinder Implemented as a Bow Rudder on a Large-Scale Ship Model

**DOI:** 10.3390/s22239137

**Published:** 2022-11-24

**Authors:** Teresa Abramowicz-Gerigk, Zbigniew Burciu

**Affiliations:** Faculty of Navigation, Gdynia Maritime University, 81-225 Gdynia, Poland

**Keywords:** rotating cylinder rudder, bow rudder, positioning system, ship maneuverability

## Abstract

This paper presents experimental studies of the force generated on the rotating cylinder implemented as a bow rudder on a large-scale ship model. The research focused on the maneuverability of the unit equipped with a rotating cylinder (RC) in the front part of the model and its future use as a steering device on small draft river barges. The study presented in this paper is a continuation of the research carried out using the small physical model of a river push train in 1:20 geometric scale equipped with two bow RCs and open water tests of separated rotating cylinders carried out in a flume tank. The experimental test setup with RC installed on the model in 1:24 geometric scale allowed to compare the parameters of standard maneuvers performed with the use of RC and without it. The proposed method based on the measurement of the ship model trajectory during maneuvers allowed to compare the hydrodynamic steering force generated by RC with the steering force generated by the conventional stern spade rudder. The results of the experiments compared with empirical models show a similar trend. RC dynamics was tested for rotational speeds up to 570 RPM (revolutions per minute) and ship model velocity up to 1 m/s. The rotating cylinder generated velocity field is presented and phenomena influencing the generated hydrodynamic force are discussed.

## 1. Introduction

Improving the maneuverability of waterborne inland units is especially important in river navigation due to the progressive reduction of maneuvering space and water depth caused by climate changes [1,2,3]. The enhancement of turning and course keeping characteristics is also important for the development of autonomous units [4,5,6]. Generally, better maneuverability of inland units has an impact on safety and reduction of costs of waterborne inland transport [7,8]. This applies in particular to river push trains with large length to depth and length to draft ratios [9,10,11].

The maneuverability of an inland unit in the given external conditions depends on its geometry, power to displacement ratio, and efficiency of control devices. The auxiliary devices applied to improve turning and course keeping characteristics are bow thrusters and passive bow rudders. 

Bow thrusters can be operated at low speeds only and their main operational disadvantage is generation of a high-energy water jet, which in limited space causes erosion of riverbed and banks and adversely affects nearby moored units. The passive bow rudders are not effective, especially at low speeds. 

A bow steering device in which the steering force is generated on a rotating cylinder—a hydrodynamic rotor—can be used in the range from slow to full ship speeds without producing the strong jets. This solution has been studied specially to improve the controllability of the push barge trains eliminating these unfavorable effects [1,2,12].

The rotating cylinder placed in the fluid flow generates Magnus force—the lifting force perpendicular to the rotor axis and flow direction. The value and direction of Magnus force depend on the fluid characteristics, flow velocity, rotational speed, and dimensions of the cylinder. However, in real conditions the rotor-generated hydrodynamic force is influenced by a number of disturbing effects, i.e., free surface effect, edge vortices formation, flow separation, and vibrations [13,14]. 

The usual methods used to predict the lift force are based on empirical equations, physical modeling [15,16,17,18], and numerical simulation [19,20,21,22]. The empirical methods are mainly developed for the large aspect ratio aerodynamic rotors operated at high Reynolds numbers [21,22]. The detailed literature study on the research related to isolated rotating cylinders was presented in [19].

The small geometric scale of the physical models and verification of numerical simulations based on model test results include uncertainties related to the limited physical and numerical modeling of flow phenomena, scale effect, and modeling of operating conditions. The large-scale model tests can bring the comparable data to verify the above-mentioned methods. 

The previous tests carried out for rotors with different height-to-diameter ratios allowed to determine the dependence of the lift force coefficient on the water inflow velocity and rotational speed of the rotor, and then to calculate the lift that can be obtained on a real object [19]. The investigations on the large-scale model presented in the paper allowed to study the rotor performance in the conditions close to the real operational conditions. The results of the selected maneuvering trials are compared with previous analysis of maneuvers performed on the push train physical model [1]. 

The steering force generated by RC causes a significant change in the reaction of the vessel compared to the traditionally used steering devices [1,2]. It is possible to determine these changes observing the ship performance during standard maneuvers [23,24]. 

The results presented in the paper led to conclusions regarding the possibility and advisability of using this type of device. They confirmed the results of earlier research on a small model and opened the possibility of building a prototype of a new bow steering device. 

The presented analysis partially supplements the lack of publications on the hydrodynamic rotor, serving as an auxiliary steering device used on shallow draft waterborne inland units, and indicates the directions of further research on the hydrodynamic force generated by the rotating cylinder under different operational conditions. The main goal will be to find the relationship between the hydrodynamic force generated by the RC and operational parameters, and to develop appropriate algorithms to control the hydrodynamic force.

## 2. Materials and Methods

The list of variables used in the paper is presented in Table 1. 

### 2.1. Experimental Test Setup

The experimental test setup was constructed in the shape of a new bow part attached to the physical manned model of 145,700 DWT tanker in 1:24 geometric scale used in professional training of marine pilots and ship masters in the Ship Handling and Training Center of the Foundation for Safety of Navigation and Environment Protection, on Lake Silm near Ilawa in Poland.

The choice of the manned model was justified due to its availability as a platform for installing a new bow with a rotating cylinder. The manned model was equipped with instruments for measuring its course, position, and data transmission from model devices. The data were transmitted and recorded by a local telemetry system.

The main particulars of the ship and physical manned model are presented in Table 2.

The new bow installed on the ship model lengthened it up to 14.40 m. The main particulars of the new bow are presented in Table 3.

The experimental test setup is presented in Figure 1.

RC was submerged together with the drive shaft, the intersection with the water surface was above the screens. The circular screens protected the cylinder against lift losses caused by edge vortices [25]. The rotor drive used in the tests had power of 1000 W, maximum rotation speed 3000 RPM. The dimensions of the tested rotors are presented in Table 4.

The program of model tests included initial trials of RC performance on the steady course and turning circle maneuvers carried out at Full Ahead and Half Ahead engine telegraph settings. 

The required maneuvering space was investigated. This space is the area needed to perform a given maneuver in the assumed external conditions. The required maneuvering space is most often described by its maximum length—ship displacement ahead—and maximum width—lateral displacement with respect to the initial position of the ship. These parameters always depend on the parameters influencing the maneuverability of the vessel: speed, settings of the steering devices, dimensions of the ship, dimensions of the area on which the maneuver is performed, and the effects of wind, waves, and current. 

The model motion parameters: course, speed, and position were measured using the onboard model devices (Figure 2) and local RTK Leica positioning system with 0.01 m accuracy. 

The trajectory of the model allowed to present the required maneuvering area necessary to perform the maneuver. 

### 2.2. Program of Model Tests

The model tests concentrated on turning circle standard maneuvers with different rudder angles performed with and without use of RC2. The rotational speed of the rotor selected on the basis of the initial trials was equal to 300 RPM. The trials were carried out at Full Ahead and Half Ahead engine telegraph settings related to the initial model speed 0.92 m/s and 0.5 m/s related to 9 knots and 4.5 knots in real scale. 

The preliminary trials included measurements of rotor generated flow velocity v_R_ at three distances l measured from the rotor axis in forward direction in bollard-pull conditions (stationary model, calm water) and trials of initial turning with different model speeds and rotor RPM carried out for RC1 and RC2 and rudder angle 0°. 

A program of the tests is presented in Table 5. 

## 3. Results

### 3.1. Rotor-Generated Velocity in Bollard-Pull Conditions

The measurements in bollard pull conditions were carried out to observe the free surface effect and RC generated flow in ahead direction at different rotational speeds. The measurements were done for RC1 at three distances l = 1 m, 0.5 m, and 0.32 m between the log and cylinder axis. 

The significant values were measured close to the cylinder surface at 500 RPM. The maximum velocities at 500 RPM are presented in Table 6. There was no flow velocity in forward direction observed for the rotational speeds less than 300 RPM.

The water level changes around the rotating cylinder caused by the pressure changes and vibrations started to appear at 400. The flow on the free surface at 500 RPM is presented in Figure 3. 

### 3.2. Steady Course Keeping Trials

The steady course keeping trials began with tests carried out at various RC rotational speeds to determine RC performance at Half Ahead and Full Ahead model speeds. There was a poor model response at low rotational speeds. At 300 RPM, the reaction of the model was comparable to its reaction for 8°–10° rudder angle. At 400 RPM, lowering of the water surface around the rotating cylinder and strong vibrations reducing the lift force appeared. To avoid these effects the rotational speed was assumed to be 300 RPM in the following trials.

Flow around RC1 at model velocities 0.5 m/s and 0.92 m/s, RC1 rotational speeds 200 RPM and 300 RPM is presented in Figure 4. 

The forward view from the model bridge during the trial is presented in Figure 5. 

The trial of steady course keeping parallel the leading lights was carried out with clockwise rotational speed of RC2 equal to 300 RPM. The stern rudder set 8°–10° to portside was used to compensate the RC generated yawing moment. The trajectory of the model (point at amidships) plotted during the trial is presented in Figure 6.

Advance and transfer are the longitudinal and transverse distances between the initial position of the point on the waterplane at the intersection of the amidships and the center plane at the beginning of the maneuver, and position of this point after completion of the maneuver.

The resultant yawing moment is zero, which means that yawing moment generated by rotating cylinder counteracts the moment generated by stern rudder. 

The model trajectory shows that it is subjected to the resultant sway force which is the sum of rudder and RC-generated forces. 

In calm weather conditions, this force caused sway of the model, observed during the trials. In strong weather conditions, the resultant sway force can counteract the sideways force of the wind.

### 3.3. Turning Circle Trials

The turning circle standard trial starts at constant ship course and speed with the ordered constant rudder angle and is finished when the course change is equal to 360°. 

The comparison of turning circle trials carried out with and without use of RC2 for the engine settings: Full Ahead (0.92 m/s) and Half Ahead (0.5 m/s), rudder angles 35°, 20°, and 10° to starboard and rotational speed of RC equal to 0 and 300 RPM clockwise are presented in Table 7 and Table 8. The standard parameters measured during maneuvers are as follows: advance A (m), transfer T (m), and tactical diameter TD (m). A and TD are the distances between the initial position of the model—point at amidships i.e., point on the waterline at the intersection of model amidships and center plain, and position after 90° and 180° course change, respectively. T is the maximum lateral displacement of this point. The parameters of the turning circle trial are presented in Figure 7.

The difference between the turning circle parameters δ (1) carried out with and without RC are presented in Table 7.
δ = (X − X_RC_)/X,(1)
where X is the turning circle parameter: A, T, TD; subscript RC means use of RC during the trial.

The biggest changes in turning were observed at small rudder angles when the small drift angle at the bow results with large lift and small drag forces on RC2.

## 4. Discussion

The biggest changes in turning due to RC operation were observed at wide turns: at small rudder angles and small drift angles at the bow, where the inflow direction from the bow was at an acute angle to the symmetry plane of the model resulting in the large lift and small drag components of the RC-generating hydrodynamic force. When a sharp turn is used as a collision avoidance maneuver, the most important is the decrease of advance. The maximum decrease of advance calculated for RC2 at Full Ahead engine setting is 25 m, in real scale (Equation (2)) it is equal to 600 m.
ΔA_real scale_ = 24 · δ_A_ · A_RC,_
(2)

The differences ΔA = δ_A_ · A_RC_ related to the ship length are presented in Table 9, they are equal from 0.3 to 1.8 ship lengths.

### 4.1. Comparison of Tactical Diameters Obtained for the Tested Model with RC and Model of a Push Train

The results were compared with the turning circle trial of the remotely controlled physical model of a 100-meter push barges train in 1:20 geometric scale described in [1,2,19], equipped with a bow steering system, consisting of two RCs. 

The advance and transfer of the turning circle performed with bow RCs and without them are presented in Figure 8. 

The research showed a significant decrease of the maneuvering space when RCs were used. The difference of TD of the turning circle performed with use of stern rudders without RCs and with them for the push train is 0.3 times the push train length. 

This difference obtained for the maneuvers presented in the paper at Full Ahead engine telegraph setting and rudder angle 35° is 0.475 times the model lengths.

### 4.2. Comparison of the Lift Forces Generated by Bow RC and Stern Rudder

The presented model tests did not allow for the measurement of the lift force generated by RC in operational conditions. However, the steady course keeping trial allowed to verify the empirical formula proposed for the isolated RC (Equation 3) [19,21].

Both the lift forces generated by RC and rudder can be estimated using lift force coefficients. The lift coefficient of the RC was obtained experimentally in [19,21], the lift coefficient of the rudder was the coefficient of the rudder profile of NACA (National Advisory Committee for Aeronautics) airfoil series. 

The estimation of the lift force generated by RC can be based on Equation (3) and empirical values of lift coefficient C_LRC_ obtained from [21].
(3)$$Y_{\mathrm{RC}}=0.5C_{\mathrm{LRC}}{\rho v}^{2}\mathrm{DH}$$
where YRC is RC generated lift force, CLRC is the lift coefficient determined from model tests of isolated rotating cylinders, ρ = 1000 kg/m3 is water density, v (m/s) is inflow velocity, D and H are the rotor height and diameter, respectively. 

This force can be compared with the force generated on the stern rudder estimated from Equation (4).
(4)$$Y_{R}=0.5C_{\mathrm{LR}}{\rho v}^{2}A_{R}$$
where Y_R_ is rudder generated lift force, C_LR_ is the lift coefficient of NACA airfoil, ρ = 1000 kg/m^3^ is water density, v (m/s) is inflow velocity, A_R_ is the rudder area. 

The C_LRC_ lift coefficient is dependent on Reynolds number (5) and rotation rate α (6).
Re = v·D/ν(5)
where Re is Reynolds number, v is inflow velocity (m/s), D is rotor diameter, and ν = 10^−6^ m^2^/s is water kinematic viscosity.
(6)$$\alpha =\frac{\mathrm{rD}}{2v}$$
where α is rotation rate, r is rotational speed (rad/s), D is rotor diameter, and v is inflow velocity (m/s).

The rotation rate calculated for RC2 rotational speed 300 RPM equal to 31.4 rad/s and inflow velocity v = 0.92 m/s is equal to α = 1.88. Reynolds number for this rotor is equal Re = 2 × 10^5^. The lift force coefficient for the rotation rate α = 1.88, is equal to 3.5 [21]. 

The lift force Y_RC_ calculated from equation (2) is equal to 196 N (2700 kN in real scale).

The lift coefficient for NACA airfoil for the rudder angles 8°–10° used during the trial is 0.6. The rudder inflow velocity can be estimated from Equation 7 [25]—efflux velocity of the propeller.
(7)$$U_{0}=c\cdot {\left(\frac{f\cdot P}{\rho \cdot D_{p}{}^{2}}\right)}^{1/3}$$
where: U_0_ is efflux velocity, c is the propeller type coefficient, f is the ratio of engine power used, P (W) is the maximum engine power, ρ (kg/m^3^) is water density, D_P_ (m) is propeller diameter.

The estimated inflow velocity for c = 1.34, f = 0.6, P = 600 W, ρ = 1000 kg/m^3^, D_P_ = 0.37 m is equal to 1.8 m/s.

The estimated rudder-generated lift force calculated from Equation 3 for C_LR_ = 0.6, inflow velocity 1.8 m/s, and A_R_ = 0.19 m^2^ is equal to 184 N (2540 kN in real scale). 

The estimated lift force generated on RC is 12 N (165 kN in real scale) greater than rudder-generated lift force.

## 5. Conclusions

Ship maneuvering performance is one of the main factors influencing safety of navigation in strong weather conditions [26]. Good maneuvering characteristics increase the possibility of collision avoidance in emergency situations [27] and decrease energy consumption due to smaller ship resistance related to ship handling [28]. 

The RC bow steering system described in the paper enables to improve vessel maneuverability, turning, and course keeping ability. The following conclusions can be drawn based on the presented model trials:there was no strong influence of free surface and the bow wave on the RC-generated steering force,RC vibrations appeared at rotational speeds greater than 400 RPM,the RC steering force depends on the drift angle at the bow,the lift force generated by the tested RC is of the same magnitude as the lift force of the stern rudder,the results of the presented research are comparable to the results obtained from model tests of 1:20 scale push train model, showing the same trend in increased controllability,the main problem with the development of the commercial application of the bow steering system is the prediction hydrodynamic force generated by the rotating cylinder in dependence on rotational speed and inflow velocity in operational conditions, necessary to control the steering force.

The installation of the bow RC was considered on river barges. The possible implementation of the device in place of the conventional bow rudder was discussed with shipowners of river barges operated in Odra river.

The presented study allowed to estimate the steering force. However, to determine the hydrodynamic characteristics of the bow RC including the hull influence, further research is necessary. 

The presented results will be followed by captive tests in the towing tank of the Maritime Advanced Research Center (CTO S.A.) to determine the hydrodynamic characteristics of the rotors. 

## Figures and Tables

**Figure 1 sensors-22-09137-f001:**
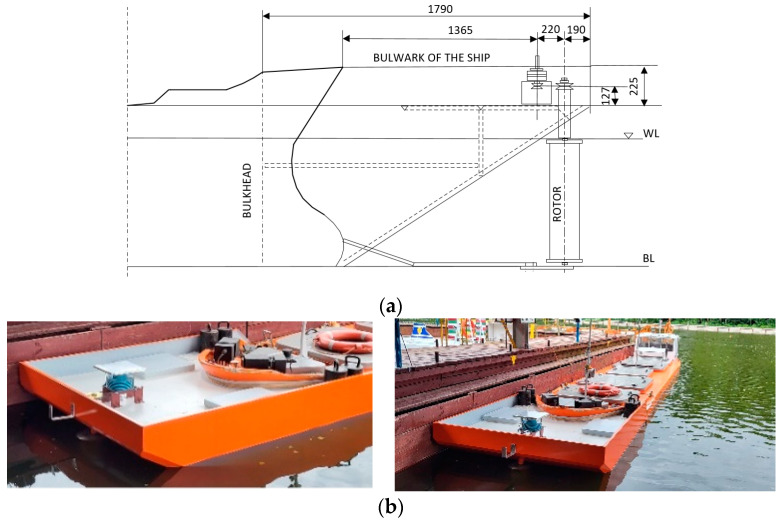
The new bow with RC attached to the ship model: (**a**) scheme of the construction of the new bow with rotor and drive; (**b**) ship model with the new bow.

**Figure 2 sensors-22-09137-f002:**
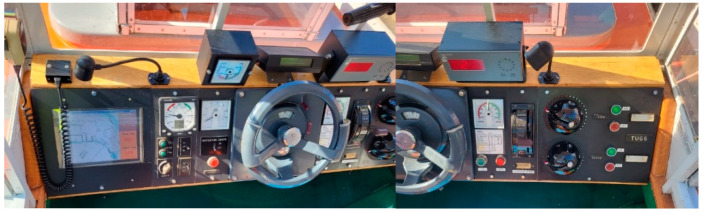
Control panel with indicators of speed, course, rate of turn, rudder angle, and engine telegraph on the model bridge.

**Figure 3 sensors-22-09137-f003:**
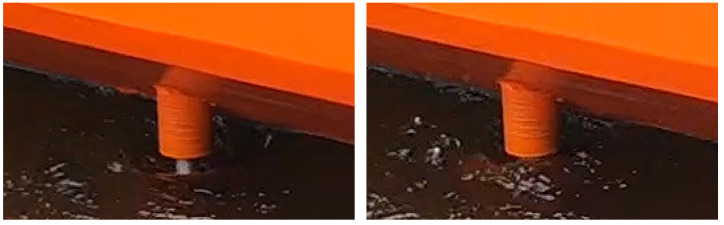
Water level changes around the rotating cylinder.

**Figure 4 sensors-22-09137-f004:**
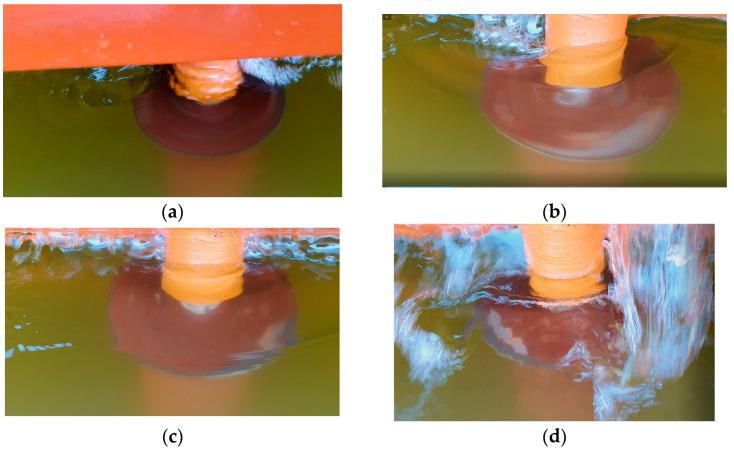
Flow around RC1: (**a**) model velocity 0.5 m/s, RC2 rotational speed 200 RPM, (**b**,**c**) model velocity 0.5 m/s, RC1 rotational speed 300 RPM, (**d**) model velocity 0.92 m/s, RC1 rotational speed 300 RPM.

**Figure 5 sensors-22-09137-f005:**
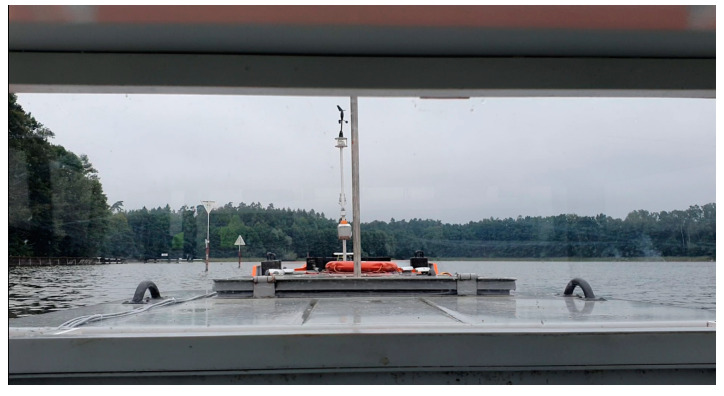
Course keeping trial parallel to the leading lights, view from the bridge of the ship model.

**Figure 6 sensors-22-09137-f006:**
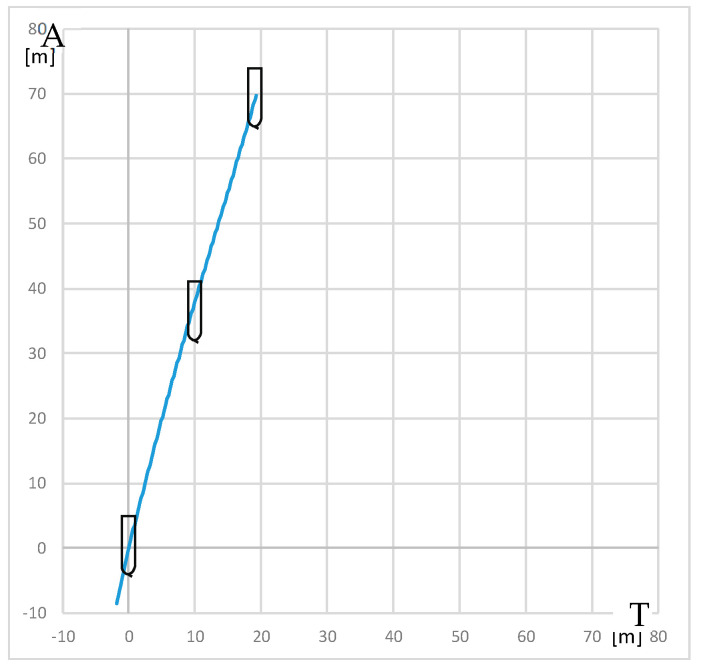
The trajectory of the model plotted during the steady course trial: A—advance, T—transfer.

**Figure 7 sensors-22-09137-f007:**
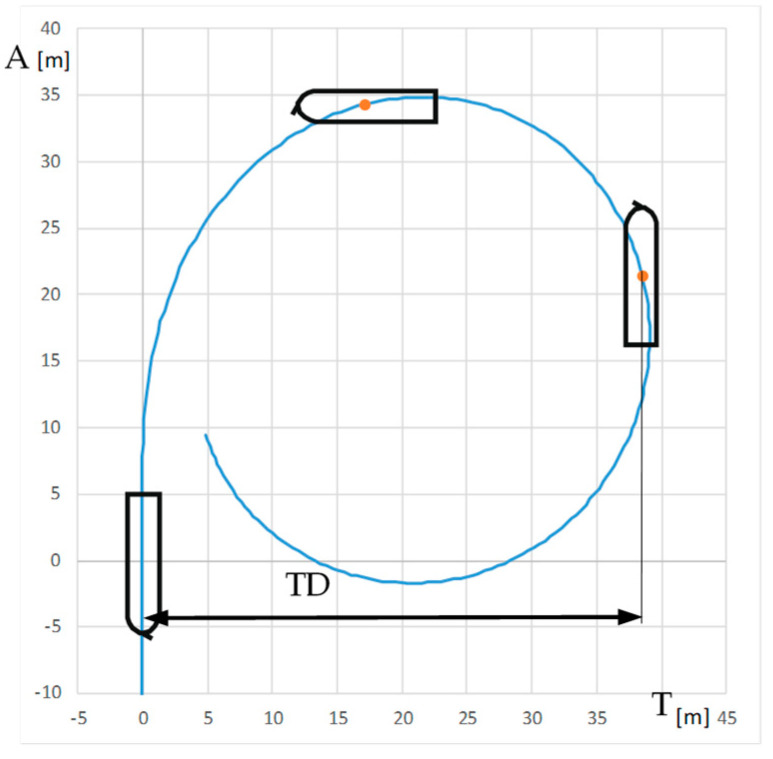
Parameters of the turning circle trial: A—advance, T—transfer, DT—tactical diameter.

**Figure 8 sensors-22-09137-f008:**
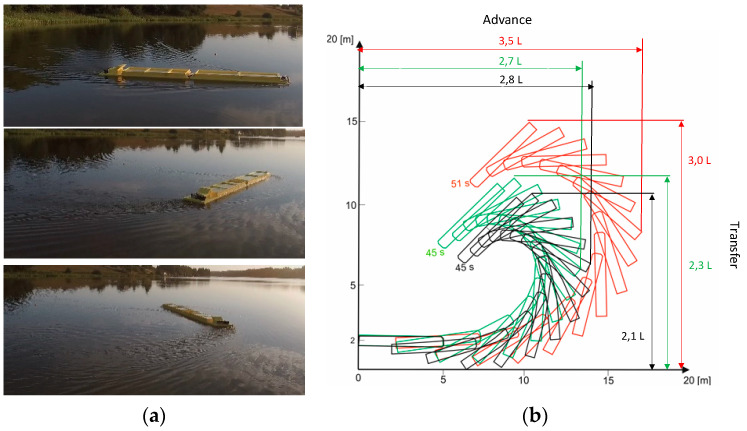
Turning circle of push train model: (**a**) turning circle trial with use of bow RCs; (**b**) maneuvering space of the model using bow RCs (red), stern rudders and bow RCs (green), stern rudders, bow RCs, and dynamical coupling system (black) [1]; L—model length.

**Table 1 sensors-22-09137-t001:** Variables used in the paper.

Parameter	Description
A (m)	advance
A_R_ (m^2^)	rudder area
B (m)	breadth
B_B_ (m)	breadth of the bow
C_LR_	lift coefficient of rudder
C_LRC_	lift coefficient of rotating cylinder
c	propeller type coefficient
D (m)	cylinder diameter
d (m)	cylinder screen diameter
D_P_ (m)	propeller diameter.
f	ratio of engine power used
H (m)	height of rotating cylinder
v_R_ (m/s)	rotor generated flow velocity
l (m)	distances from the rotor in forward direction
L_OA_ (m)	length over all
L_B_ (m)	length of the bow
N_RC_ (W)	rotating cylinder drive power
P (W)	engine power
r (rad/s)	rotational speed
Re	Reynolds number
T_B_ (m)	draft of the bow
TD (m)	tactical diameter
U_0_ (m/s)	efflux velocity
v (m/s)	inflow velocity
Y_R_ (N)	rudder generated lift force
Y_RC_ (N)	rotor generated lift force
α	rotation rate
ΔA_real scale_ (m)	change in advance due to RC operation
δ	difference between turning circle trial parameters
ν (m^2^/s)	kinematic viscosity
ρ (kg/m^3^)	water density

**Table 2 sensors-22-09137-t002:** Main particulars of the ship and physical manned model.

Parameter	Ship	Model
L_OA_ (m)	292.90	12.20
B (m)	48.00	2.00
T (m)	15.33	0.64

**Table 3 sensors-22-09137-t003:** Main particulars of the bow of the physical manned model.

Bow parameter	Value
L_B_ (m)	2.20
B_B_ (m)	2.00
T_B_ (m)	0.64

**Table 4 sensors-22-09137-t004:** Parameters of the tested rotors.

Parameter	RC1	RC2
H (m)	0.60	0.60
D (m)	0.22	0.11
d (m)	0.30	0.19
r (RPM)	0–570	0–570
N_RC_ (W)	1000	1000

**Table 5 sensors-22-09137-t005:** Program of model tests. Tests in bollard-pull conditions. Tests at Full Ahead model speed: turning circle and steady course keeping trials. Tests at Half Ahead model speed: turning circle trial.

Bollard-Pull v_R_ Measurement		Full Ahead				Half Ahead Turning Circle	
		Turning Circle		Steady Course			
l (m)	RC1 RPM	Ruder Angle	RC2 RPM	Ruder Angle	RC2 RPM	Ruder Angle	RC2 RPM
1	0–570	35°	0	8°–10°	300	35°	0
		20°	0			20°	0
0.5		10°	0			10°	0
		0°	300			0°	300
0.32		35°	300			35°	300
		20°	300			20°	300
		10°	300			10°	300

**Table 6 sensors-22-09137-t006:** RC—generated flow velocity in forward direction at 500 RPM.

l (m)	v_R_ (m/s)
1	0.080
0.5	0.094
0.32	0.990

**Table 7 sensors-22-09137-t007:** Turning circle trial at Full Ahead and Half Ahead engine settings, rudder angles set to starboard, and clockwise rotation of RC2.

Ruder Angle	RC2 RPM	Full Ahead				Half Ahead			
		Turning Circle	A	T	TD	Turning Circle	A	T	TD
35°	0	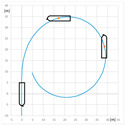	34	17	38	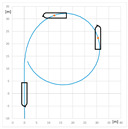	31	12	30
20°	0	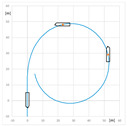	48	23	52	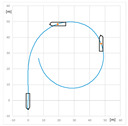	48	19	47
10°	0	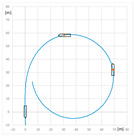	58	30	68	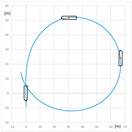	51	30	66
0°	300	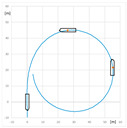	44	26	55	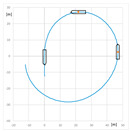	27	21	46
35°	300	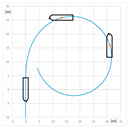	25	13	31	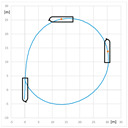	25	13	30
20°	300	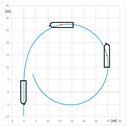	27	15	34	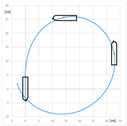	25	14	32
10°	300	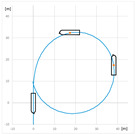	32	17	38	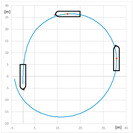	26	19	40

**Table 8 sensors-22-09137-t008:** Difference between turning circle parameters for two engine settings: Full Ahead and Half Ahead, rudder angles set starboard, and clockwise rotational speeds of RC2 trial.

Rudder Angle	δ (%)					
	Full Ahead			Half Ahead		
	A	T	TD	A	T	TD
35°	26	24	18	19	−8	0
20°	44	35	35	48	26	32
10°	45	43	44	49	37	39

**Table 9 sensors-22-09137-t009:** Nondimensional advance in turning circle trial for engine setting Full Ahead and Half Ahead, rudder angles set to starboard, and clockwise direction of RC2 rotational speeds RPM.

Rudder Angle	Full Ahead		Half Ahead	
	A/L	ΔA/L	A/L	ΔA/L
35°	2.4	0.6	1.7	0.3
20°	3.3	1.5	1.9	0.9
10°	4.0	1.8	2.2	1.1

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
