# Peer review of "Investigations of Hydrodynamic Force Generated on the Rotating Cylinder Implemented as a Bow Rudder on a Large-Scale Ship Model"

_sensors, 2022, doi:10.3390/s22239137_

Round 1

Reviewer 1 Report

The paper presents experimental studies of the force generated on the rotating cylinder implemented as a bow rudder on a large-scale ship model. As claimed by the authors, the paper is a continuation of the research carried out using the small physical model of a river push train. The topic is interesting. The paper's presentation, however, is poor. Results of the research are rarely discussed. Additionally, the reference writing does not adhere to the established standard format. Below are my comments:

1. What is the justification of the ship and physical manned model as presented in Table 1?

2. Line 159- Why did the steady-state trials reveal the best rotor performance at 300 RPM? Factors influencing this decision should be discussed.

3. Discussions on experimental results need to be compared with similar research results from other researchers

4. In Section 4.2, the comparison of the lift forces generated by bow RC and stern rudder was presented. However, the philosophy of the importance of comparing lift forces is not explained in detail. The author only presents the calculation results and compares them with each other.

5. Line 209. It is strongly suggested to add the discussion about the trajectory of the model. Also, please revise Figure 6.  The description of x and y axis are missing.

6. Please add nomenclature. There are some parameters that are not defined.

Author Response

Dear Reviewer 1,

Thank you for your comments. They help to improve the paper a lot. We can agree with all of them. The following corrections are made in the paper.

  1. What is the justification of the ship and physical manned model as presented in Table 1?

The following justification is added before Table 1

“The choice of the manned model was justified due to its availability as a platform for installing a new bow with a rotating cylinder. The manned model was equipped with instruments for measuring its course, position and data transmission from model devices. The data were transmitted and recorded by a local telemetry system.”

  1. Line 159- Why did the steady-state trials reveal the best rotor performance at 300 RPM? Factors influencing this decision should be discussed.

The following explanation is added at the beginning of paragraph 3.2:

“The steady course keeping trials began with tests carried out at various RC rotational speeds to determine RC performance at Half Ahead and Full Ahead model speeds. There was a poor model response at low rotational speeds. At 300 rpm the reaction of the model was comparable to its reaction for 8°-10° rudder angle. At 400 RPM, lowering of the water surface around the rotating cylinder and strong vibrations reducing the lift force appeared. To avoid these effects the rotational speed was assumed to be 300 RPM in the following trials.”

  1. Discussions on experimental results need to be compared with similar research results from other researchers

The authors did not find the similar research described in the literature. Most experiments are related to the isolated rotating cylinders, without influence of free surface effect. The detailed literature study on the research related to isolated rotating cylinders was presented in our previous paper published in Sensors.

  1. Abramowicz-Gerigk, T.; Burciu, Z.; Jachowski, J.; Kreft, O.; Majewski, D.; Stachurska, B.; Sulisz, W.; Szmytkiewicz, P. Ex-perimental Method for the Measurements and Numerical Investigations of Force Generated on the Rotating Cylinder under Water Flow. Sensors 2021, 21, 2216. https://doi.org/10.3390/s21062216.

The following sentence is added in Introduction (line 57):

 “ The detailed literature study on the research related to isolated rotating cylinders was presented in [19].”

Therefore experimental results of trials with RC were compared with results of trials using the conventional rudder. The lift force generated by the rotating cylinder was calculated using empirical formula with lift coefficient for the similar rotating cylinder available in literature:

  1. Karabelas, S.J.; Koumroglou, B.C.; Argyropoulos, C.D.; Markatos, N.C. High Reynolds number turbulent flow past a rotating cylinder. Applied Mathematical Modelling 2012, 36, 379–398..
  2. In Section 4.2, the comparison of the lift forces generated by bow RC and stern rudder was presented. However, the philosophy of the importance of comparing lift forces is not explained in detail. The author only presents the calculation results and compares them with each other.

The explanation was presented in section 3.2 as the answer to Comment 5 and following explanation is added in section 4.2:

“The presented model tests did not allow for the measurement of the lift force generated by RC in operational conditions. However, the steady course keeping trial allowed to verify the empirical formula proposed for the isolated RC (Equation 3) [19,21].”

  1. Line 209. It is strongly suggested to add the discussion about the trajectory of the model. Also, please revise Figure 6.  The description of x and y axis are missing.

The description of x and y axes and discussion about the trajectory are added:

“Advance and transfer are the longitudinal and transverse distances between the initial position of the point on the waterplane at the intersection of the amidships and the centre plane at the beginning of the manoeuvre, and position of this point after completion of the manoeuvre.

The resultant yawing moment is zero what means that yawing moment generated by rotating cylinder counteracts the moment generated by stern rudder.

The model trajectory shows that it is subjected to the resultant sway force which is the sum of rudder and RC – generated forces.

In calm weather conditions this force caused sway of the model, observed during the trials. In strong weather conditions the resultant sway force can counteract the sideways force of the wind.”

  1. Please add nomenclature. There are some parameters that are not defined.

The nomenclature was added in Table 1

Reviewer 2 Report

The paper presents experimental studies of the force generated on the rotating cylinder implemented as a bow rudder on a large-scale ship model. However, there are some problems which need to be revised carefully.

1.      The whole manuscript shows an experimental setup and gives some results. However, theoretical descriptions and analyses are much less so that the novelty of manuscript is hard to be shown. Moreover, the theoretical depth is not enough.

2.      In the experiment part, the comparison results between proposed method and other classical methods are not provided.

Author Response

Dear Reviewer 2

Thank you for your comments which help us to improve the paper. We can agree with them. Please find our explanation and corrections made in the text.

The classical methods described in the literature are related to the isolated rotating cylinders, without influence of free surface effect. The detailed literature study on the research related to isolated rotating cylinders was presented in our previous paper published in Sensors:

  1. Abramowicz-Gerigk, T.; Burciu, Z.; Jachowski, J.; Kreft, O.; Majewski, D.; Stachurska, B.; Sulisz, W.; Szmytkiewicz, P. Ex-perimental Method for the Measurements and Numerical Investigations of Force Generated on the Rotating Cylinder under Water Flow. Sensors 2021, 21, 2216. https://doi.org/10.3390/s21062216.

The following sentence is added in Introduction (line 57):

  The detailed literature study on the research related to isolated rotating cylinders was presented in [19].

The experimental results of trials with RC were compared with results of trials using the conventional rudder and lift force calculated using empirical formula with lift coefficient for the similar rotating cylinder available in literature:

  1. Karabelas, S.J.; Koumroglou, B.C.; Argyropoulos, C.D.; Markatos, N.C. High Reynolds number turbulent flow past a rotating cylinder. Applied Mathematical Modelling 2012, 36, 379–398.

The explanation of comparison with classical method is added in line 279”:

“The presented model tests did not allow for the measurement of the lift force generated by RC in operational conditions. However, the steady course keeping trial allowed to verify the empirical formula proposed for the isolated RC (Equation 3) [19,21].

Reviewer 3 Report

The paper topic is interesting, results based on calculations, I suggest to improuve the conclusions, if possible, with more informations about the possible installation on the real scale unit. 

Author Response

Dear Reviewer 3

Thank you for your comment. The following correction is made in the paper:

“The installation of the bow RC was considered on river barges. The possible implementation of the device in in place of the conventional bow rudder was discussed with shipowners of river barges operated in Odra river.”

Reviewer 4 Report

In general, the paper is interesting and suitable for the Sensors journal. However, there are several obvious minor errors. Thus, the paper quality needs to be enhanced.  Some comments are given below.  

Comments:

§  Abstract: double check the sentence: “The motor-generated velocity field in bollard-pull condition”.

§  Abstract: improve the sentence “The proposed method based on the measurement of the ship model position allowed to compare the hydrodynamic steering force generated on RC with the steering force generated on the conventional stern spade rudder.”

§  Abstract: what is “rpm”?

§  Line 47: fix “depends on”.

§  Line 48: change “However” to “However, ”

§  Line 49: change “i.e.” to “ i.e.,”

§  The literature review is not enough.  The authors should cite more recent relevant papers especially published in the MDPI journals such as Sensors, Applied Sciences, Machines and JMSE.

§  Line 70: double check “allowed to”.

§  Improve the format of Table 1.

§  Line 136: “Table 2” should be corrected to “Table 4”.

§  The caption of Table 4 is not specific.

§  Line 183: Table 3 or Table 5? Double check the table numbering errors.

§  Lines 186-188: notations should be italic. Double check the places.

§  Line 191: an error (191) is shown in the first row of Table 3.

§  Line 304: change “The results” to “the results”.

§  Line 315, what is “(CTO S.A.)”?

Author Response

Dear Reviewer 4,

Thank you for your remarks and comments. They help a lot to improve the paper We can agree with all of them. The following corrections are made in the paper:

  • Abstract: double check the sentence: “The motor-generated velocity field in bollard-pull condition”.

This sentence is removed from the text

  • Abstract: improve the sentence “The proposed method based on the measurement of the ship model position allowed to compare the hydrodynamic steering force generated on RC with the steering force generated on the conventional stern spade rudder.”

The sentence is changed to:

The proposed method based on the measurement of the ship model trajectory during manoeuvres allowed to compare the hydrodynamic steering force generated by RC with the steering force generated by the conventional stern spade rudder.”

  • Abstract: what is “rpm”?

rpm is revolutions per minute, it is changed to “RPM”

  • Line 47: fix “depends on”.

It is changed to “depend on”

  • Line 48: change “However” to “However, ”

It is changed to “However,”

  • Line 49: change “i.e.” to “ i.e.,”

It is changed to “i.e.:”

  • The literature review is not enough.  The authors should cite more recent relevant papers especially published in the MDPI journals such as Sensors, Applied Sciences, Machines and JMSE.

The three recent papers on ship manoeuvring and power-propulsion performance are added:

  1. Sutulo, S.; Guedes Soares, C. Review on Ship Manoeuvrability Criteria and Standards. J. Mar. Sci. Eng. 2021, 9, 904. https://doi.org/10.3390/jmse9080904
  2. Zhao, X.; He, Y.; Huang, L.; Mou, J.; Zhang, K.; Liu, X. Intelligent Collision Avoidance Method for Ships Based on COLRGEs and Improved Velocity Obstacle Algorithm. Appl. Sci. 2022, 12, 8926. https://doi.org/10.3390/app12188926
  3. Lebkowski, A.; Wnorowski, J. A Comparative Analysis of Energy Consumption by Conventional and Anchor Based Dynamic Positioning of Ship. Energies 2021, 14, 524. https://doi.org/10.3390/en14030524

  • Line 70: double check “allowed to”.

It is changed to “led to “

  • Improve the format of Table 1.

The format of Table 1 is improved

  • Line 136: “Table 2” should be corrected to “Table 4”.

It is changed to “Table 4”

  • The caption of Table 4 is not specific.

The Table 4 caption is changed to the following:

Table 4. Program of model tests. Tests in bollard-pull conditions. Turning circle and steady course keeping tests at Full Ahead model speed. Turning circle tests at Half Ahead model speed.

  • Line 183: Table 3 or Table 5? Double check the table numbering errors.

Table numbering is corrected and changed again due to introduction of new Table 1 with the list of variables.

  • Lines 186-188: notations should be italic. Double check the places.

In template from 2022 notations style is normal.

  • Line 191: an error (191) is shown in the first row of Table 3.

Table 3 is changed to Table 7

  • Line 304: change “The results” to “the results”.

It is changed

  • Line 315, what is “(CTO S.A.)”?

It is  Maritime Advanced Research Centre (CTO S.A.) - this explanation is included in the text.

Round 2

Reviewer 1 Report

Some issues have been added by the authors. Now, it can be acceptable

Author Response

Dear Reviewer,

Thank you for your comment, we really appreciate your help in improving the paper.

Reviewer 4 Report

In general, it seems that the authors did not carefully improve the paper according to our comments. There are still some flaws in the revised paper.  Thus, I have to suggest the authors should take more time and efforts to improve the paper.  Some comments are given below. 

Comments:

1)      The full term of “RPM” should be given in Abstract.

2)      It is not good to put Table 1 (variables/parameters?) in Introduction.

3)      The Introduction section should be improved by emphasizing motivation and significance of this study.

4)      Literature review is still not enriched. Why the new references [26-29] are added in Section 5?

5)      Notations should be italic.

6)      The format of tables should be further improved.

7)      There are still some language errors, e.g., fix a typo “manoeurability” on Line 377; change “,” to “;” on Lines 379-391; fix “…force, however to determine…” on Line 396; etc.

Author Response

Dear Reviewer,

Thank you for your comments, we really appreciate your help in improving the paper. We agree with all your comments. Please find our corrections.

  • The full term of “RPM” should be given in Abstract.

The full term “(revolutions per minute)” is given in Abstract

  • It is not good to put Table 1 (variables/parameters?) in Introduction.

The list of variables is moved to section 2

  • The Introduction section should be improved by emphasizing motivation and significance of this study.

The motivation and significance is emphasised in Introduction:

Line 73

“The results presented in the paper led to conclusions regarding the possibility and advisability of using this type of device. They confirmed the results of earlier research on a small model and opened the possibility of building a prototype of a new bow steering device.”

Line 79

The main goal will be to find the relationship between the hydrodynamic force generated by the RC and operational parameters, and to develop appropriate algorithms to control the hydrodynamic force.

  • Literature review is still not enriched. Why the new references [26-29] are added in Section 5?
  • The literature review is included in our previous paper:

[19] Abramowicz-Gerigk, T.; Burciu, Z.; Jachowski, J.; Kreft, O.; Majewski, D.; Stachurska, B.; Sulisz, W.; Szmytkiewicz, P. Experimental Method for the Measurements and Numerical Investigations of Force Generated on the Rotating Cylinder under Water Flow. Sensors 2021, 21, 2216. https://doi.org/10.3390/s21062216.

This reference is cited in Introduction:

Line 55

The detailed literature study on the research related to isolated rotating cylinders was presented in [19].

  • The new references are related to operational aspects of RC applications therefore they are cited in Conclusions.
  • Notations should be italic.

In the 2022 template the style of Equations is normal  

  • The format of tables should be further improved.

Format of tables is improved

  • There are still some language errors, e.g., fix a typo “manoeurability” on Line 377; change “,” to “;” on Lines 379-391; fix “…force, however to determine…” on Line 396; etc.

The language errors are corrected